# A lightweight Transformer guided by features from multiple receptive fields for few-shot fine-grained image classification

## Abstract

Convolutional neural networks (CNNs) and vision Transformers (ViTs) play key roles in few-shot fine-grained image classification (FSFGIC). One of the main challenges of FSFGIC is how to consistently learn high-quality feature representations from different very limited fine-grained datasets. CNNs struggle with long-range dependencies due to their inherent localized receptive fields, and ViTs might impair high-frequency information, e.g., local texture information. Furthermore, ViTs require a large number of training samples to infer feature properties such as translation invariance, locality, and the hierarchy of visual data, while FSFGIC's training samples are extremely limited. To address the problems mentioned, a new lightweight Transformer guided by features from multiple receptive fields (LT-FMRF) is proposed which has considered how to manage long-range dependencies and how to extract local features with multiple scales, global features, and fused features from input images for increasing inter-class differences and consistently obtaining high-quality feature representations from different types of limited training datasets. Furthermore, the proposed LT-FMRF can be easily embedded into a given few-shot episodic training mechanism for end-to-end training from scratch. Experimental results conducted on five widely used FSFGIC datasets consistently show significant improvements over twenty state-of-the-art end-to-end training-based methods.

## 1 Introduction

Few-shot fine-grained image classification (FSFGIC) (Zhang et al., 2024) aims to use a limited number of training samples to train a network for accurately classifying images (e.g., with birds (Wah et al., 2011)) belonging to subordinate object categories of the same entry-level category. Existing FSFGIC methods can be roughly classified into two groups (Zhang et al., 2021): meta learning-based and metric learning-based FSFGIC methods. Meta learning-based methods aim to enable a given model to learn quickly and obtain good generalization performance when adapting to new tasks in a scenario with limited data by optimizing model parameters or learning strategies. Metric learning-based methods use metric functions such as cosine distance or Euclidean distance to determine the category of samples based on the similarity between different samples.

Convolutional neural networks (CNNs) and vision Transformers (ViTs) play a key role in FSFGIC. Although CNNs have the capability to obtain local feature information from images well, they have difficulties handling long-range dependencies due to their inherent localised receptive fields. Transformers have the capability to effectively capture low-frequency signals from images, i.e., global feature information (e.g., global shapes and structures). However, Transformers (Park & Kim, 2022) might also impair high-frequency information (e.g., local textures information) to a certain extent. Currently, pre-trained Transformers or some branches of Transformers (e.g., encoder or decoder) have been widely employed in FSFGIC for improving classification accuracy. It was indicated in (Liu et al., 2021) that existing pre-trained Transformer models (Sun et al., 2022; Dosovitskiy et al., 2021) for FSFGIC require significantly more training data compared to CNNs to infer feature properties such as translation invariance, locality, and the hierarchy of visual data, for FSFGIC tasks where the training samples are extremely limited. These give us the following inspiration: With limited

training samples and in a given few-shot scenario for a training mechanism, how to train a lightweight Transformer model end-to-end from scratch to perform FSFGIC tasks well?

In this work, a new lightweight Transformer network (LT-Net) is proposed for addressing the aforementioned problems. The designed LT-Net in this paper is mainly based on the following two considerations: (1) Vision Transformers (ViTs) have the capability to capture dependencies between image patches and capture long-range global pattern information well. However, due to the lack of convolutional inductive biases (Liu et al., 2021), ViTs rely more on large-scale image datasets than CNNs. (2) In the human visual system (Bullier, 2001; Kauffmann et al., 2014), information from different frequency bands is indispensable and is fused in some ways to extract more important and unique features. In this manner, a new lightweight Transformer guided by features from multiple receptive fields (LT-FMRF) is proposed which has considered how to manage long-range dependencies and how to extract local features with multiple scales, global features, and fused features from input images for increasing inter-class differences, and consistently obtaining high-quality feature representations from different training datasets. Furthermore, with limited training samples, the proposed LT-FMRF can be easily embedded into a given few-shot episodic training mechanism for end-to-end training from scratch. Extensive experiments on five benchmark datasets (i.e., CUB-200-2011 (Wah et al., 2011), Stanford Dogs (Khosla et al., 2011), Stanford Cars (Krause et al., 2013), meta-iNat (Van Horn et al., 2018; Wertheimer & Hariharan, 2019), and tiered meta-iNat dataset (Van Horn et al., 2018; Wertheimer & Hariharan, 2019)) demonstrate the effectiveness and superiority of the proposed LT-FMRF over twenty state-of-the-art end-to-end training-based methods.

## 2 RELATED WORK

This section outlines several existing approaches which relate to the proposed method.

### 2.1 META LEARNING-BASED FSFGIC

The core concept of meta-learning is "learning to learn". Attention mechanisms were widely used in meta learning-based FSFGIC methods which aim to learn salient feature representations from input images. In the work of (Ruan et al., 2021), a spatial attention comparison network was proposed which contains a feature selective comparison module to fuse multi-scale feature maps of support and query images by arranging different weights pixel by pixel. Wang et al. (2024) introduced a dual-channel attention meta-learning architecture which contains an embedding module and a feature calibration module for addressing the problem that traditional FSFGIC methods indiscriminately obtain semantic feature information from each part of the input image.

Alternately, feature alignment techniques have been proposed, which aim to align the spatial locations of objects between support images and query images to capture subtle differences. In (Song et al., 2024), a feature disentanglement alignment network was introduced that aims to enhance the model's generalizability by maintaining the consistency of extraneous features throughout the fine-tuning process. In the work of (Leng et al., 2024), a feature distribution alignment architecture was proposed, which takes into account the differences in feature distributions between tasks that are ignored by the current meta-learning based methods, and then the Kullback-Leibler divergence method is used to improve the similarity of the extracted feature distributions.

In addition to the above methods, knowledge distillation techniques were proposed to improve learning efficiency and generalization ability by transferring knowledge from complex models to simplified models. In (Wu et al., 2023), a task-specific meta-distillation was presented in which the teacher and student models are trained simultaneously in the procedure of meta-learning. Then in the validation process, the teacher model is first fine-tuned, and the adjusted teacher model guides the adjustment of the student model. Peng et al. (2024) proposed a semantic-guided visual adaptation architecture which intends to extend the vision-language pre-trained model by integrating implicit knowledge distillation, vision-specific contrastive loss, and cross-modal contrastive loss to generate discriminative and adaptive visual features.

## 2.2 METRIC LEARNING-BASED FSFGIC

Attention mechanisms were also widely employed in metric learning-based FSFGIC algorithms which have the capability to provide effective feature weighting strategies for metric learning, enabling a model to process complex data more effectively, thereby improving the classification performance and generalization ability of the model. In (Tang et al., 2022), used a combination of multi-scale feature pyramids and multi-level attention pyramids to enhance the internal representation of features and reduce the uncertainty caused by the background mediated by limited samples. In (Li et al., 2023b), visual self-attention mechanisms are used to to infer local feature relationships, model spatial long-distance dependencies, estimate representative prototypes, and develop discriminative prototype-query pairs.

Feature alignment techniques have been introduced for learning feature embedding. A background suppression and foreground alignment network (Zha et al., 2023) was presented which aims to suppress the background content of images and align the foreground of support and query image pairs. Ma et al. (2024) proposed a cross-layer and cross-sample feature optimization network (C2-Net) which integrates feature maps from multiple network layers and improves the matching results between query features and support samples by adjusting the query features from both channel and spatial perspectives.

Feature reconstruction techniques have also been widely applied in metric learning-based FSFGIC tasks. In (Wertheimer et al., 2021), a feature map reconstruction network (FRN) was introduced which reconstructs query features directly from support features by ridge regression in closed form. Li et al. (2024) argued that existing reconstruction methods do not address the overfitting problem due to the scarcity of samples during training. In Li et al. (2024), a self-reconstruction metric module and a constrained cross-entropy loss based on FRN (Wertheimer et al., 2021) were proposed. In the work of (Li et al., 2023a), a local content enrichment cross reconstruction network (LCCRN) was proposed, in which a local content enrichment module was designed to learn discriminative local feature representations and a cross reconstruction module was introduced to combine these local features with the appearance details obtained from a separate embedding module to enhance the semantic understanding of the network.

## 2.3 TRANSFORMERS FOR FSFGIC

In (Wang et al., 2023), the encoder, decoder, and cross-attention in the Transformer architecture (Vaswani et al., 2017) were utilized to model the support image representation, query image representation, and metric learning for performing different FSFGIC tasks. Huang & Choi (2023) presented a self-attention-based prototype enhancement network that integrates discriminative features with channel information to obtain representative class prototypes for addressing feature redundancy in prototype networks. In (Wu et al., 2024), a self-attention module (Vaswani et al., 2017) was employed to reconstruct the query set from the support set for increasing inter-class difference and the support set was reconstructed from the query set for reducing intra-class difference.

Alternatively, pre-trained Transformers were also widely utilized in FSFGIC. In the work of (Sun et al., 2022), global and local feature interaction based on a pre-trained vision Transformer, named GL-ViT, was proposed to mine few-shot feature attributes. To address the overfitting problem caused by insufficient data, He et al. (2022) introduced a hierarchical cascaded Transformer that leverages intrinsic image structures through spectral token pooling and optimizes learnable parameters via latent attribute surrogates. Hao et al. (2023) proposed a class-aware patch embedding adaptation (CPEA) network which aims to remove the noise of single-label annotations and avoid supervision collapse without aligning semantically related regions.

## 3 PROPOSED METHOD

In this section, we first formulate the definition of FSFGIC and then detail a lightweight Transformer guided by features from multiple receptive fields.

### 3.1 PROBLEM STATEMENT

A typical FSFGIC setting contains a support set $S$ and a query set $Q$. Support set $S$ contains $\mathcal{N}$ different image classes and each class in $\mathcal{N}$ is composed of $\mathcal{K}$ labeled samples. Query set $Q$ is composed of unlabeled samples. Set $S$ and set $Q$ have the same data-label space. The goal of FSFGIC is to train a model which has the ability to classify each query sample $q$ ($q \in Q$) into its corresponding class in $\mathcal{N}$. Thus, the FSFGIC task is called a $\mathcal{N}$-way $\mathcal{K}$-shot task (Vinyals et al., 2016).

### 3.2 THE PROPOSED LIGHTWEIGHT TRANSFORMER GUIDED BY FEATURES FROM MULTIPLE RECEPTIVE FIELDS (LT-FMRF)

The architecture of the designed LT-FMRF in this paper is shown in Fig. 1. The details of LT-FMRF will be illustrated as follows.

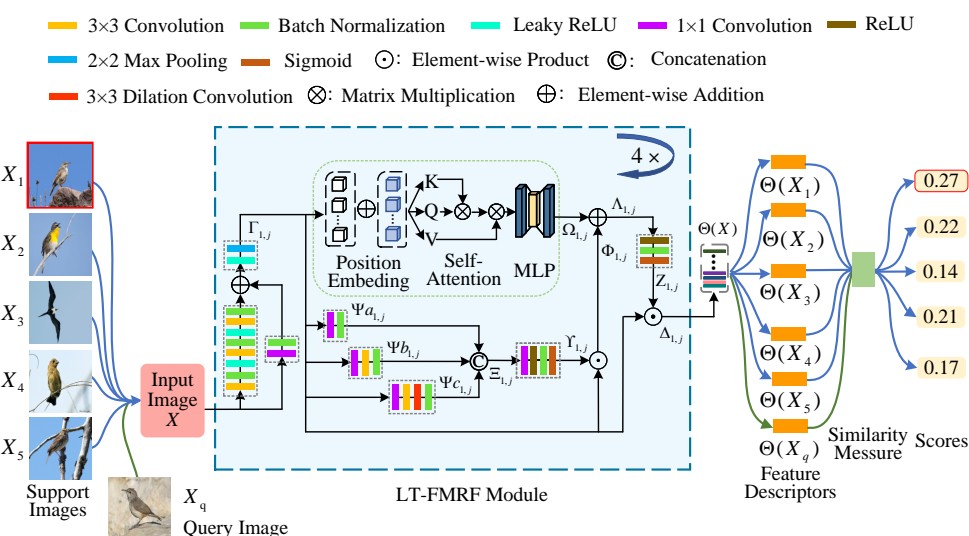

Figure 1: The pipeline of proposed LT-FMRF for a 5-way 1-shot FSFGIC task.

Due to the lack of convolutional inductive bias (Liu et al., 2021), ViTs rely more on large-scale image datasets than CNNs, which is obviously not desirable on FSFGIC. In this way, a convolutional neural network module of ResNet-12 (Lee et al., 2019) or Conv-4 (Wertheimer et al., 2021) is used to obtain initial feature tensors from an input image $X$ ($X \in S \cup Q$) with a size of $N \times N$. After passing through the first convolutional module of ResNet-12, 64 initial feature tensors $\Gamma_{1,j}$ ($j=1, ..., 64$) with a size of $\frac{N}{2} \times \frac{N}{2}$ can be obtained. Then the initial feature tensors $\Gamma_{1,j}$ ($j=1, ..., 64$) are fed into the ViT module to obtain self-attention based feature tensors (SA-FTs) $\Omega_{1,j}$ ($j=1, ..., 64$) with a size of $\frac{N}{2} \times \frac{N}{2}$. The purpose of this is to maintain convolutional inductive bias, reduce the number of training samples required for ViT training itself, and have the capability to deal with long-range dependencies properly.

Furthermore, inspired by (Bullier, 2001) and (Kauffmann et al., 2014) that information in different frequency bands is indispensable in the human visual system and will be fused in some way to extract more important and unique features, three different convolutional modules with different filtering kernel sizes (i.e., 1×1 and 3×3) are employed to smooth the initial feature tensors $\Gamma_{1,j}$ ($j=1, ..., 64$) for obtaining multiple receptive fields based feature tensors $\Psi a_{1,j}$, $\Psi b_{1,j}$, and $\Psi c_{1,j}$ ($j=1, ..., 64$) with a size of $\frac{N}{2} \times \frac{N}{2}$. Then the Concat($\cdot$) function is utilized to concatenate feature representations $\Psi a_{1,j}$, $\Psi b_{1,j}$, and $\Psi c_{1,j}$ as follows:

$$\Xi_{1,j} = \text{Concat}(\Psi a_{1,j}, \Psi b_{1,j}, \Psi c_{1,j}) \in R^{(3 \times 64) \times \frac{N}{2} \times \frac{N}{2}}, \tag{1}$$

where $R$ represents real space. In order to imitate human vision to fuse features of different bands for extracting unique features, 1×1 convolution operation is employed to fuse multiple scale features

$\Xi_{1,j}$. Providing that the two initial feature tensors, e.g., $\check{\Gamma}_a$ and $\check{\Gamma}_b$ of two categories of images (e.g., $X_a$ and $X_b$) obtained from the convolutional neural network module of ResNet-12 (Lee et al., 2019) as shown in Fig. 1 have Gaussian distributions with zero mean and standard deviations $\omega_a$ and $\omega_b$ (here $\omega_a > \omega_b$), the difference $D$ between the two initial feature tensors can be obtained (Zhang et al., 2024) by their corresponding standard deviations, i.e., $D = \omega_a - \omega_b$. After passing through the three scale feature fusion, the fused feature tensors from two categories of images ($X_a$ and $X_b$) have Gaussian distributions with zero mean and standard deviations $\sqrt{3}\omega_a$ and $\sqrt{3}\omega_b$. In this way, the difference $D$ between the two feature descriptors of the two categories of images is $\sqrt{3}(\omega_a - \omega_b)$. It is clear that multi-scale feature fusion can not only obtain unique feature information, but also effectively amplify the differences between feature information of different categories. Furthermore, we found that under the condition of limited training samples in FSFGIC, it is not true that the deeper the network, the better the classification performance (see ablation study). The ReLU, Batch Normalization, and Sigmoid operations are performed on the fused feature tensors for obtaining the weights of the fused feature tensors $\Upsilon_{1,j}$ ($j = 1, ..., 64$) with a size of $\frac{N}{2} \times \frac{N}{2}$. Then element-wise product operation is employed to multiply the weight tensors $\Upsilon_{1,j}$ and the initial feature tensors $\Gamma_{1,j}$ for obtaining multiple receptive field based feature tensors (MRF-FTs) $\Phi_{1,j}$ ($j = 1, ..., 64$) with a size of $\frac{N}{2} \times \frac{N}{2}$.

It is worth noting that the self-attention based feature tensor (SA-FTs) $\Omega_{1,j}$ can handle long-range dependencies well, but they may not have the capability to handle local texture information Park & Kim (2022). In this work, element-wise addition operation is used to fuse SA-FTs and MRF-FTs for obtaining fused feature tensors $\Lambda_{1,j}$ ($j = 1, ..., 64$) with a size of $\frac{N}{2} \times \frac{N}{2}$. The fused feature tensors $\Lambda_{1,j}$ contain global and local feature information, have the capability to handle long-range dependencies, and can effectively amplify the differences between feature information of different categories. In order to avoid overfitting caused by increasing network depth, the ReLU, Batch Normalization, and Sigmoid operations are performed on the fused feature tensors $\Lambda_{1,j}$ for obtaining the weight tensors $Z_{1,j}$ ($j = 1, ..., 64$) with a size of $\frac{N}{2} \times \frac{N}{2}$. Then element-wise product operation is employed to multiply the weight tensors $Z_{1,j}$ and the initial feature tensors $\Gamma_{1,j}$ for obtaining self-attention and multiple receptive field based feature tensors (SA-MRF-FTs) $\Delta_{1,j}$ ($j = 1, ..., 64$) with a size of $\frac{N}{2} \times \frac{N}{2}$.

The designed LT-FMRF network contains four modules. Then the self-attention and multiple receptive field based feature tensors (SA-MRF-FTs) $\Delta_{1,j}$ will be sent into the second, the third, and the fourth modules for obtaining feature representations $\Theta(X)$. It is worth to note that if ResNet-12 (Lee et al., 2019) is used for obtaining the initial feature tensors, the sizes of the input image and the sizes of the output feature tensors by the four modules are the same as the ResNet-12, which are $3 \times 84 \times 84$, $64 \times 42 \times 42$, $160 \times 21 \times 21$, $320 \times 10 \times 10$, and $640 \times 5 \times 5$ where the first number represents the number of channels and the second and third numbers represent the length and width of the feature map respectively. If Conv-4 (Wertheimer et al., 2021) is used for obtaining the extracted initial feature tensors, the sizes of the input image and the sizes of the output feature tensors by the four modules are the same as Conv-4, which are $3 \times 84 \times 84$, $64 \times 42 \times 42$, $64 \times 21 \times 21$, $64 \times 10 \times 10$, and $64 \times 5 \times 5$.

Feature representations of support images of each class obtained from LT-FMRF are denoted as $\Theta_{S_c}$. Meanwhile, feature representations of a query image obtained from LT-FMRF are denoted as $\Theta_Q$. Then the support set of each class are reconstructed into a query image using the closed solution (Wertheimer et al., 2021) based on LT-FMRF. The goal of feature reconstruction on LT-FMRF is to find matrices $W_Q$ such that $W_Q \cdot \Theta_{S_c} \approx \Theta_Q$. Finding the optimal solution is equivalent to solving a least squares problem as follows:

$$\overline{W}_Q = \underset{W_Q}{\arg\min} ||\Theta_Q - W_Q \cdot \Theta_{S_c}||^2 + \lambda ||W_Q||^2 \tag{2}$$

where $|| \cdot ||$ is the Frobenius norm, $\lambda$ is the regularization parameter, and $\overline{W}_Q$ is the optimal weight matrices for reconstructing query images $\overline{Q}_{Rc}$ as follows:

$$\begin{aligned} \overline{W}_Q &= \Theta_Q \cdot \Theta_{S_c}^T (\Theta_{S_c} \cdot \Theta_{S_c}^T + \lambda I)^{-1}, \\ \overline{\Theta}_{Q_c} &= \overline{W}_Q \cdot \Theta_{S_c}. \end{aligned} \tag{3}$$

For a given class $c$, the Euclidean metric is utilized to compute the distance from query images $\Theta_Q$ to the reconstructed query images $\overline{\Theta}_{Q_c}$ as follows:

$$M = \frac{1}{r}||\Theta_Q - \overline{\Theta}_{Q_c}||^2. \tag{4}$$

The final predicted probability is defined as follows:

$$M_z = \frac{e^{-\mu M}}{\sum_{c=1}^{C} e^{-\mu M}}. \tag{5}$$

Here $z$ represents the $z$th query image, and $\mu$ is a learnable weight parameter. Finally the stochastic gradient descent optimization (Bottou, 2010) with a cross-entropy loss is used to train the whole network for performing FSFGIC tasks.

## 4 EXPERIMENT

### 4.1 DATASETS

The proposed LT-FMRF network is evaluated on five fine-grained image datasets: CUB-200-2011 (Wah et al., 2011), Stanford Cars (Krause et al., 2013), Stanford Dogs (Khosla et al., 2011), meta-iNat (Van Horn et al., 2018; Wertheimer & Hariharan, 2019), and tiered meta-iNat (Van Horn et al., 2018; Wertheimer & Hariharan, 2019). The CUB-200-2011 (Wah et al., 2011) dataset contains 200 bird classes with 11,788 samples. The Stanford Cars dataset is composed of 196 car classes with 16,185 samples. The Standford Dogs dataset contains 120 dog classes with 20,580 samples. The meta-iNat dataset consists of 1,135 wildlife categories. The tiered meta-iNat dataset is a variant of meta-iNat that introduces a larger domain gap between training and testing classes. For fair comparisons, we follow the data splits described in (Ma et al., 2024) which are shown in Table 1.

Table 1: The class split of the five fine-grained datasets. $N_{\text{train}}$, $N_{\text{val}}$, and $N_{\text{test}}$ are the numbers of classes in the auxiliary set, validation set, and test set respectively.

| Dataset | $N_{\text{train}}$ | $N_{\text{val}}$ | $N_{\text{test}}$ |
|---|---|---|---|
| CUB-200-2011 | 100 | 50 | 50 |
| Stanford Cars | 130 | 17 | 49 |
| Stanford Dogs | 70 | 20 | 30 |
| meta-iNat | 908 | - | 227 |
| tiered meta-iNat | 781 | - | 354 |

### 4.2 IMPLEMENTATION DETAILS

We conduct experiments in the 5-way 1-shot and 5-way 5-shot FSFGIC settings on the above five datasets. All experiments in this work are conducted using the PyTorch framework on 2 NVIDIA 3090 Ti GPUs through data parallelism. ResNet-12 and Conv-4 are selected as the backbones for obtaining feature representations using stochastic gradient descent (Bottou, 2010) with a cross-entropy loss. The initial learning rate was 0.1, with weight decay set to 5e-4. The learning rate was reduced to 0.01 after 400 iterations. We employed standard data augmentation techniques, including random crop, random horizontal flip, and color jitter, to enhance training stability. For all experiments, this paper validates the average accuracy of 10,000 randomly generated tasks for obtaining the top-1 mean classification accuracy results under the standard 5-way 1-shot and 5-way 5-shot settings. Meanwhile, the 95% confidence intervals are obtained and reported.

### 4.3 PERFORMANCE COMPARISON

In this part, the classification performance of the proposed LT-FMRF is compared with twenty state-of-the-art methods (i.e., ProtoNet (Snell et al., 2017), DN4 (Li et al., 2019), DSN (Simon et al., 2020), BSNet (Li et al., 2021), VFD (Xu et al., 2021), FRN+TDM (Lee et al., 2022), DeepEMD (Zhang et al., 2023), LRPABN (Huang et al., 2021b), BiFRN (Wu et al., 2024), OLSA (Wu et al., 2021), HelixFormer (Zhang et al., 2022), C2-Net (Ma et al., 2024), LMPNet (Huang et al., 2021a), DAN (Xu

et al., 2022), DeepBDC (Xie et al., 2022), BSFANet (Zha et al., 2023), SRNet (Li et al., 2024), PaCL (Wang et al., 2022), LCCRN (Li et al., 2023a), and FRN (Wertheimer et al., 2021)). The experimental results on the CUB-200-2010, Stanford Dogs, Stanford Cars, meta-iNat, and tiered meta-iNat datasets are summarized in Table 2 and Table 3. The results associated with the method marked by the † tag are derived from our implementation of the open-source code, conducted under the same experimental conditions.

It can be observed from Table 2 and Table 3 that the performance of our proposed LT-FMRF is significantly better than baseline methods on the CUB-200-2011, Stanford Dogs, meta-iNat, and tiered meta-iNat datasets. For the Stanford Cars dataset, the proposed method achieves the best and fourth best performances based on Conv-4 and ResNet-12 on the 5-way 1-shot classification task, while our method achieves the best performance on the 5-way 5-shot task. These results demonstrate the effectiveness of the proposed LT-MRFF network. Take the 5-way 1-shot and 5-way 5-shot FSFGIC tasks based on Conv-4 on the Stanford Dogs dataset as an example, compared with ProtoNet, DN4, DSN, BSNet, VFD, LRPABN, FRN, FRN+TDM, PaCL, DAN, DeepEMD, BiFRN, and C2-Net, our proposed LT-MRFF achieves 24.23%, 31.51%, 26.37%, 27.47%, 13.86%, 25.17%, 10.36%, 8.12%, 11.13%, 11.08%, 24.16%, 6.15%, and 4.47% improvements for the 5-way 1-shot results and 14.84%, 15.8%, 26.19%, 13.71%, 12.61%, 24.76%, 6.42%, 5.9%, 8.11%, 8.42%, 19.87%, 4.32%, and 4.38% improvements for the 5-way 5-shot results.

Table 2: Comparison results of different methods on the CUB-200-2011, Stanford Dogs, and Stanford Cars datasets under two different backbones (methods labeled by † denote our implementations). The best performance is indicated in bold.

| Backbone | Method | CUB-200-2011 | | Stanford Dogs | | Stanford Cars | |
|---|---|---|---|---|---|---|---|
| | | 5-way Accuracy (%) | | | | | |
| | | 1-shot | 5-shot | 1-shot | 5-shot | 1-shot | 5-shot |
| Conv-4 | ProtoNet (Snell et al., 2017) | 64.82±0.23 | 85.74±0.14 | 46.66±0.21 | 70.77±0.16 | 50.88±0.23 | 74.89±0.18 |
| | DN4 (Li et al., 2019) | 57.45±0.89 | 84.41±0.58 | 39.08±0.76 | 69.81±0.69 | 34.12±0.68 | 87.47±0.47 |
| | DSN (Simon et al., 2020) | 72.56±0.92 | 84.62±0.60 | 44.52±0.82 | 59.42±0.71 | 53.45±0.86 | 65.19±0.75 |
| | BSNet (Li et al., 2021) | 62.84±0.95 | 85.39±0.56 | 43.42±0.86 | 71.90±0.68 | 40.89±0.77 | 86.88±0.50 |
| | VFD (Xu et al., 2021) | 68.42±0.92 | 82.42±0.61 | 57.03±0.86 | 73.00±0.66 | - | - |
| | LRPABN (Huang et al., 2021b) | 63.63±0.77 | 76.06±0.58 | 45.72±0.75 | 60.94±0.66 | 60.28±0.76 | 73.29±0.58 |
| | FRN† (Wertheimer et al., 2021) | 73.73±0.21 | 88.46±0.13 | 60.53±0.21 | 79.19±0.15 | 67.48±0.22 | 87.24±0.12 |
| | FRN+TDM (Lee et al., 2022) | 74.39±0.21 | 88.89±0.11 | 62.77±0.22 | 79.71±0.14 | 72.26±0.21 | 89.55±0.10 |
| | PaCL (Wang et al., 2022) | 74.07±0.70 | 88.75±0.38 | 59.76±0.70 | 77.50±0.48 | 72.21±0.68 | 88.02±0.36 |
| | DAN (Xu et al., 2022) | 72.89±0.50 | 86.60±0.31 | 59.81±0.50 | 77.19±0.35 | 70.21±0.50 | 85.55±0.31 |
| | DeepEMD (Zhang et al., 2023) | 64.08±0.50 | 80.55±0.71 | 46.73±0.49 | 65.74±0.63 | 61.63±0.27 | 72.95±0.38 |
| | LCCRN (Li et al., 2023a) | 76.22±0.21 | 89.39±0.13 | - | - | 71.62±0.21 | 86.41±0.12 |
| | BiFRN† (Wu et al., 2024) | 76.39±0.20 | 90.61±0.11 | 64.66±0.22 | 81.27±0.14 | 75.33±0.20 | 90.91±0.10 |
| | C2-Net† (Ma et al., 2024) | 78.63±0.46 | 89.48±0.26 | 69.81±0.50 | 84.39±0.29 | 79.52±0.45 | 91.15±0.24 |
| | Ours | **81.07±0.19** | **92.64±0.10** | **70.89±0.22** | **85.61±0.13** | **80.62±0.20** | **94.77±0.07** |
| ResNet-12 | LMPNet (Huang et al., 2021a) | - | - | 61.89 | 68.21 | 68.31 | 80.27 |
| | OLSA (Wu et al., 2021) | - | - | 64.15±0.49 | 78.28±0.32 | 77.03±0.46 | 88.85±0.46 |
| | FRN† (Wertheimer et al., 2021) | 82.86±0.19 | 92.41±0.11 | 76.76±0.21 | 88.74±0.12 | 86.90±0.17 | 95.69±0.07 |
| | FRN+TDM (Lee et al., 2022) | 83.26±0.20 | 92.80±0.11 | 75.98±0.22 | 88.70±0.13 | 86.91±0.17 | 96.11±0.07 |
| | HelixFormer (Zhang et al., 2022) | 81.66±0.30 | 91.83±0.17 | 65.92±0.49 | 80.65±0.36 | 79.40±0.43 | 92.26±0.15 |
| | DeepBDC (Xie et al., 2022) | 81.98 ± 0.44 | 92.24 ± 0.24 | 73.57 ± 0.46 | 86.61 ± 0.27 | 82.28 ± 0.42 | 93.51 ± 0.20 |
| | DeepEMD (Zhang et al., 2023) | 75.59±0.30 | 88.23±0.18 | 70.38±0.30 | 85.24±0.18 | 80.62±0.26 | 92.63±0.13 |
| | LCCRN (Li et al., 2023a) | 82.97±0.19 | 93.63±0.10 | - | - | 87.04±0.17 | 96.19±0.07 |
| | BSFANet (Zha et al., 2023) | 82.27±0.46 | 90.76±0.26 | 69.58±0.50 | 82.59±0.33 | 88.93±0.38 | 95.20±0.20 |
| | SRNet (Li et al., 2024) | 83.82±0.18 | 93.45±0.10 | 76.54±0.21 | 88.52±0.12 | 88.02±0.16 | 96.23±0.07 |
| | BiFRN† (Wu et al., 2024) | 82.03±0.19 | 92.78±0.10 | 77.40±0.21 | 88.41±0.12 | **90.28±0.14** | 97.26±0.05 |
| | C2-Net† (Ma et al., 2024) | 83.65±0.20 | 92.57±0.10 | 77.72±0.46 | 89.59±0.24 | 86.48±0.40 | 94.07±0.22 |
| | Ours | **84.39±0.19** | **94.25±0.09** | **77.84±0.21** | **89.79±0.11** | 87.32±0.17 | **97.41±0.05** |

Furthermore, take six images as shown in Fig. 2(a) as an example, the model attention region visualization technique based on the gradient-weighted class activation mapping (Grad-CAM) (Selvaraju et al., 2017) on ResNet-12 is utilized to illustrate the advantage of the proposed LT-FMRF. In Grad-CAM, regions with higher energies represent more discriminative parts of the image. The attention maps of the six images of FRN (Wertheimer et al., 2021) and the proposed LT-FMRF are shown in Fig. 2(b)

Table 3: Comparison results of different methods on the meta-iNat and tiered meta-iNat datasets under the Conv-4 backbone. The best performance is indicated in bold.

| Method | meta-iNat | | tiered meta-iNat | |
|---|---|---|---|---|
| | 5-way Accuracy (%) | | | |
| | 1-shot | 5-shot | 1-shot | 5-shot |
| ProtoNet (Snell et al., 2017) | 53.78 | 73.80 | 35.47 | 54.85 |
| Covar.pool (Wertheimer & Hariharan, 2019) | 57.15 | 77.20 | 36.06 | 57.48 |
| DN4 (Li et al., 2019) | 62.32 | 79.76 | 43.82 | 64.17 |
| DSN (Simon et al., 2020) | 58.08 | 77.38 | 36.82 | 60.11 |
| CTX (Doersch et al., 2020) | 60.03 | 78.80 | 36.83 | 60.84 |
| FRN (Wertheimer et al., 2021) | 61.98 | 80.04 | 43.95 | 63.45 |
| FRN+TDM (Lee et al., 2022) | 63.97 | 81.60 | 44.05 | 62.91 |
| DeepEMD (Zhang et al., 2023) | 54.48 | 68.36 | 36.05 | 48.55 |
| MCL (Liu et al., 2022) | 64.66 | 81.31 | 44.08 | 64.61 |
| C2-Net (Ma et al., 2024) | 71.47 | 85.47 | 49.04 | 67.25 |
| Ours | **72.13** | **87.14** | **51.27** | **72.34** |

and (c) respectively. It can be observed from Fig. 2(b) and (c) that compared with FRN, the proposed LT-FMRF has the capability to better focus on the classification targets themselves. Furthermore, take the 5-way 1-shot FSFGIC task on the CUB-200-2011 dataset as an example, the loss curves of FRN and the proposed LT-FMRF during training and validation stages are shown in Fig. 3(a) and (b), and the accuracy curves of FRN and the proposed LT-MRFF during training and validation stages are shown in Fig. 3(c) and (d). It can be observed from Fig. 3 that compared with FRN, the proposed LT-FMRF has a lower loss and a better accuracy.

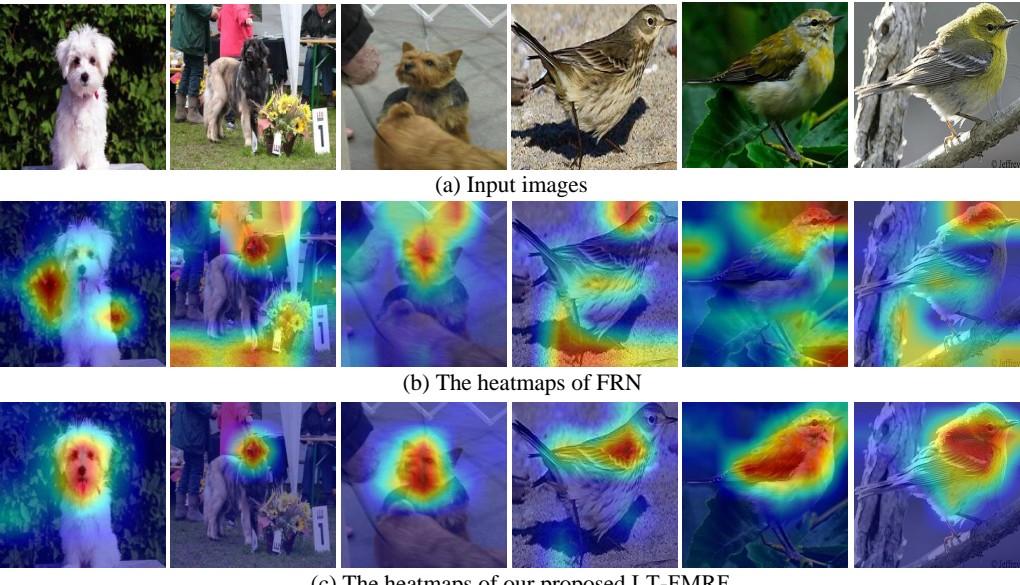

(a) Input images

(b) The heatmaps of FRN

(c) The heatmaps of our proposed LT-FMRF

Figure 2: The heatmaps of six images visualized by the FRN and the proposed LT-FMRF.

## 4.4 ABLATION STUDIES

To further study the sensitivity of our approach, ablation experiments are conducted as follows.

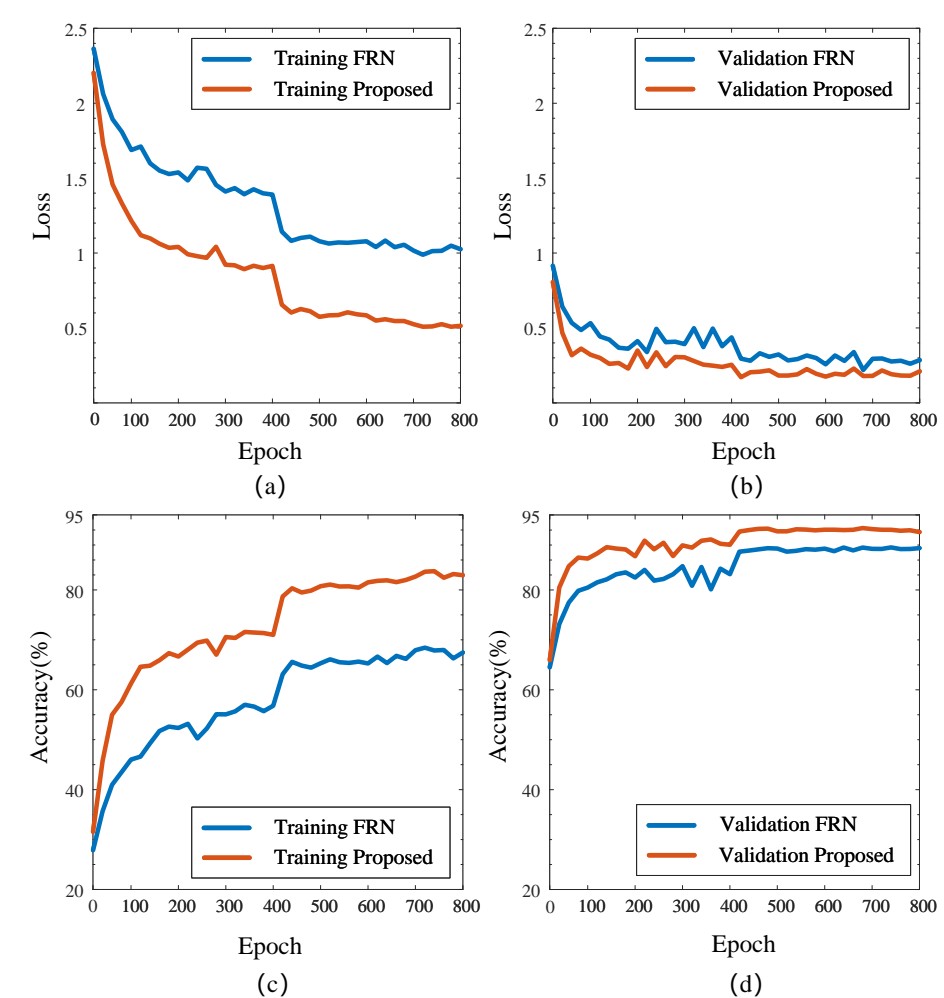

Figure 3: Examples of the loss and accuracy curves of FRN and the proposed method for the 5-way1-shot FSFGIC task on the CUB-200-2011 dataset. The loss curves of FRN and the proposed method during training and validation are shown in (a) and (b); The accuracy curves of FRN and the proposed method during training and validation are shown in (c) and(d).

**Influence of SA-FTs on FSFGIC.** Experiments are conducted for investigating the effect of the self-attention feature tensors (SA-FTs) of our proposed LT-FMRF on performance. Based on the designed LT-FMRF strategy with the designed multiple receptive fields module, the proposed LT-FMRF with and without the SA-FTs are employed to perform 5-way 1-shot and 5-way 5-shot FSFGIC tasks on the CUB-200-2011, Stanford Dogs, and Stanford Cars datasets. The results on the 5-way 1-shot and 5-way 5-shot tasks based on the Conv-4 and ResNet-12 backones are shown in Table 4. It is observed from Table 4 that the proposed LT-FMRF strategy with SA-FTs achieves the overall best classification performance.

**Influence of MRF-FTs on FSFGIC.** Experiments are conducted for investigating the effect of the multiple receptive field feature tensors (MRF-FTs) of our proposed LT-FMRF on performance. Based on the designed LT-FMRF strategy with the designed self-attention feature tensors (SA-FTs), the proposed LT-FMRF with and without MRF-FTs are employed to perform 5-way 1-shot and 5-way 5-shot FSFGIC tasks on the CUB-200-2011, Stanford Dogs, and Stanford Cars datasets. The results on the 5-way 1-shot and 5-way 5-shot tasks based on the Conv-4 and ResNet-12 backones are shown in Table 4. It is observed from Table 4 that the proposed LT-FMRF strategy with MRF-FTs achieves the overall best classification performance.

**Influence of the different numbers of LT-FMRF modules.** We also investigate the effect of the different numbers of LT-FMRF modules by performing 5-way 1-shot and 5-way 5-shot FSFGIC tasks

Table 4: Ablation experiments using only SA-FTs or MRF-FTs.

| Backbone | SA-FTs | MRF-FTs | CUB-200-2011 | | Stanford Dogs | | Stanford Cars | |
| --- | --- | --- | --- | --- | --- | --- | --- | --- |
| | | | 5-way Accuracy (%) | | | | | |
| | | | 1-shot | 5-shot | 1-shot | 5-shot | 1-shot | 5-shot |
| Conv-4 | × | × | 73.73±0.21 | 88.46±0.13 | 60.53±0.21 | 79.29±0.15 | 67.48±0.22 | 87.24±0.12 |
| | ✓ | × | 79.93±0.20 | 91.77±0.11 | 68.33±0.22 | 83.21±0.13 | 75.94±0.21 | 91.00±0.10 |
| | × | ✓ | 79.04±0.20 | 91.30±0.11 | 67.85±0.22 | 83.23±0.14 | 79.01±0.20 | 94.21±0.08 |
| | ✓ | ✓ | **81.07±0.19** | **92.64±0.10** | **70.89±0.22** | **85.61±0.13** | **80.62±0.20** | **94.77±0.07** |
| ResNet-12 | × | × | 82.86±0.19 | 92.41±0.10 | 76.76±0.21 | 88.74±0.12 | 86.90±0.17 | 95.69±0.07 |
| | ✓ | × | 83.82±0.19 | 93.90±0.10 | 77.76±0.21 | 89.25±0.11 | 87.01±0.18 | 97.21±0.06 |
| | × | ✓ | 83.89±0.19 | 93.40±0.10 | **78.01±0.21** | 89.04±0.12 | **87.73±0.17** | 97.28±0.06 |
| | ✓ | ✓ | **84.39±0.19** | **94.25±0.09** | 77.84±0.21 | **89.79±0.11** | 87.32±0.17 | **97.41±0.05** |

on the CUB-200-2011, Stanford Dogs, and Stanford Cars datasets. It is worth to note that when the number of LT-FMRF modules is set to five, the 2×2 max pooling layer of the fifth module in Conv-4 and ResNet-12 is removed, and its corresponding size of the output feature tensors in Conv-4 and ResNet-12 is 64×5×5 and 640×5×5 respectively. The results on the 5-way 1-shot and 5-way 5-shot tasks based on the Conv-4 and ResNet-12 backbones with different numbers of LT-FMRF modules are summarized in Table 5. It is observed from Table 5 that the proposed method with four LT-FMRF modules achieves the overall best classification performance. Therefore, the proposed method with four LT-FMRF modules is recommended for our designed architecture.

Table 5: The impact of the different numbers of LT-FMRF modules tested on the CUB-200-2011, Stanford Dogs, and Standford Cars datasets.

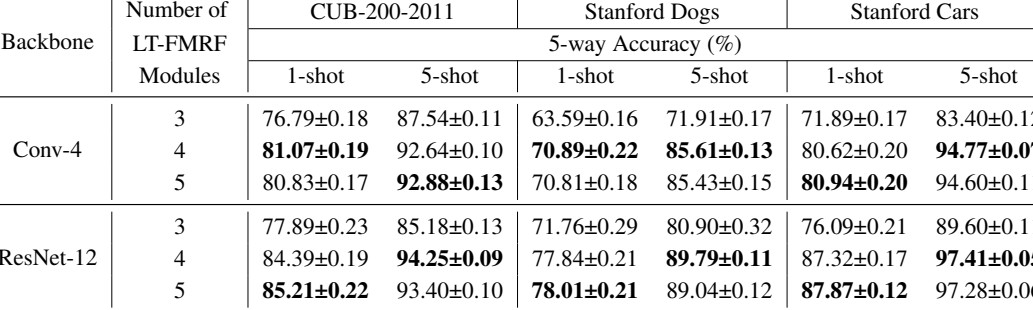

| Backbone | Number of LT-FMRF Modules | CUB-200-2011 | | Stanford Dogs | | Stanford Cars | |
| --- | --- | --- | --- | --- | --- | --- | --- |
| | | 5-way Accuracy (%) | | | | | |
| | | 1-shot | 5-shot | 1-shot | 5-shot | 1-shot | 5-shot |
| Conv-4 | 3 | 76.79±0.18 | 87.54±0.11 | 63.59±0.16 | 71.91±0.17 | 71.89±0.17 | 83.40±0.12 |
| | 4 | **81.07±0.19** | 92.64±0.10 | **70.89±0.22** | **85.61±0.13** | 80.62±0.20 | **94.77±0.07** |
| | 5 | 80.83±0.17 | **92.88±0.13** | 70.81±0.18 | 85.43±0.15 | **80.94±0.20** | 94.60±0.11 |
| ResNet-12 | 3 | 77.89±0.23 | 85.18±0.13 | 71.76±0.29 | 80.90±0.32 | 76.09±0.21 | 89.60±0.11 |
| | 4 | 84.39±0.19 | **94.25±0.09** | 77.84±0.21 | **89.79±0.11** | 87.32±0.17 | **97.41±0.05** |
| | 5 | **85.21±0.22** | 93.40±0.10 | **78.01±0.21** | 89.04±0.12 | **87.87±0.12** | 97.28±0.06 |

Overall, these ablation studies confirm that both SA-FTs and MRF-FTs are essential components of the LT-FMRF framework. Together, they effectively capture both global and local feature information, manage long-range dependencies, and amplify the differences between different types of features, thereby improving the model's ability to tackle the challenges of FSFGIC tasks.

## 5  CONCLUSION

In this paper, a new lightweight Transformer guided by features from multiple receptive fields (LT-FMRF) is proposed for FSFGIC. The designed LT-FMRF has the capability to manage long-range dependencies and extract local features with multiple scales, global features, and fused features from input images for increasing inter-class differences and consistently obtaining high-quality feature representations from different types of limited training datasets. Furthermore, the proposed LT-FMRF can be easily embedded into any given few-shot episodic training mechanisms for end-to-end training from scratch. Experimental results conducted on five widely used FSFGIC datasets consistently show significant improvements over twenty state-of-the-art end-to-end training-based methods.

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
