# OpenReview forum: "A lightweight Transformer guided by features from multiple receptive fields for few-shot fine-grained image classification"
_ICLR.cc/2025/Conference — ICLR 2025 Conference Withdrawn Submission_

### Official Review · Reviewer_ocDB · 2024-10-25

**Soundness:** 2
**Presentation:** 1
**Contribution:** 1
**Rating:** 3
**Confidence:** 5

**Summary:**

This paper presents a lightweight module called LT-FMRF, which uses convolutional layers with varying kernel sizes to manage long-range dependencies and extract multi-scale local features, global features, and fused features from input images. This approach aims to enhance inter-class differences and achieve high-quality feature representations. Experimental results across five datasets demonstrate its effectiveness.

**Strengths:**

1.The paper is easy to follow.

2.The experiment results looks good but it is not clear whether the pre-trained models were used or not.

**Weaknesses:**

1.This paper designs a very simple lightweight module, LT-FMRF, which uses convolutional layers with multiple kernel sizes to smooth initial features. However, using convolutional kernels of varying sizes to capture multi-scale features is a well-established approach (such as [1-4] and so on) that has been extensively explored in numerous models and visual tasks. Additionally, ResNet-12 and Conv-4 serve as the basic backbones for this task, and all compared methods utilize these architectures. As a result, the paper lacks novelty and significant contributions.

[1] Jiang W, Huang K, Geng J, et al. Multi-scale metric learning for few-shot learning[J]. IEEE Transactions on Circuits and Systems for Video Technology, 2020, 31(3): 1091-1102.

[2] Han M, Wang R, Yang J, et al. Multi-scale feature network for few-shot learning[J]. Multimedia Tools and Applications, 2020, 79: 11617-11637.

[3] Askari F, Fateh A, Mohammadi M R. Enhancing Few-Shot Image Classification through Learnable Multi-Scale Embedding and Attention Mechanisms[J]. arXiv preprint arXiv:2409.07989, 2024.

[4] Zhang Y, Sidibé D, Morel O, et al. Multiscale attention-based prototypical network for few-shot semantic segmentation[C]//2020 25th International Conference on Pattern Recognition (ICPR). IEEE, 2021: 7372-7378.



2.The author does not provide incremental experimental results for the proposed module, leaving it unclear how its accuracy compares to other existing public models. Consequently, the effectiveness of the proposed module cannot be truly verified.

3.The author claims that the LT-FMRF module enhances inter-class differences; however, they provide no factual evidence, quantitative results, or any analysis to support this assertion.

4. The paper lacks sufficient internal ablation studies (components of convolution layer of LT-FMRF) to truly validate the effectiveness of the proposed module LT-FMRF.

5.Figure 3 compares the training process of this method with a very low-performing approach of FRN (with an accuracy difference of over 10%), which does not provide meaningful insights. A comparison should be made with similar or incremental models instead. In addition, the heatmap in Figure 2 is unconvincing in its comparison with FRN. A more effective approach would be to compare the LRF method with the LRF + LT-FMRF method, rather than contrasting the proposed method with a significantly lower-accuracy baseline.

6.Table 5 presents experiments on the number of LT-FMRF modules but does not analyze why using more modules results in better performance for 1-shot results while leading to lower accuracy for 5-shot results. Furthermore, the paper contains several grammatical errors.

**Questions:**

Overall, this paper shows limited innovation and contribution, insufficient experimentation, and lacks theoretical analysis to support the authors' claims.

---

### Official Review · Reviewer_3wMw · 2024-10-30

**Soundness:** 2
**Presentation:** 2
**Contribution:** 1
**Rating:** 3
**Confidence:** 5

**Summary:**

The paper introduces a lightweight Transformer model, called LT-FMRF, designed to improve FSFGIC. By combining features from multiple receptive fields with self-attention, LT-FMRF aims to handle both local and global dependencies, which helps address the challenge of limited training data typical in few-shot scenarios. Experimental results across five benchmark datasets show that LT-FMRF achieves better accuracy than several state-of-the-art methods.

**Strengths:**

The paper is well-organized, presenting a clear problem statement and relevant background on FSFGIC. The experimental results and tables are well-structured, aiding in understanding the model's comparative strengths. Some technical details in the LT-FMRF architecture and fusion mechanisms may require additional clarification, but the paper generally succeeds in communicating the key ideas and findings.

**Weaknesses:**

In general, the motivation for this submission is easy to understand, but the novelty is limited. In addition, there are still several weaknesses, as follows:
1. The proposed method largely combines existing techniques, such as integrating CNN and Transformer layers, multiple receptive fields, and self-attention mechanisms. For similarity measurement, it uses feature reconstruction, as in FRN. While these choices are effective, the approach introduces limited novelty in both network architecture and similarity measurement compared to recent state-of-the-art methods, especially those ViTs-based methods specifically designed for FSL.

2. As a researcher in FSL, I find the motivations behind this submission unclear. The paper proposes a lightweight Transformer model for few-shot fine-grained image classification (FSFGIC), aiming to learn high-quality feature representations from limited data. However, this goal aligns with general few-shot learning objectives and does not address the unique characteristics of fine-grained images. In particular, the design lacks specific adaptations for fine-grained classification. Highlighting how the proposed modules differ from previous designs specifically in handling fine-grained distinctions would strengthen the paper.

3. The paper explains its multi-receptive field and self-attention approaches for enhancing feature extraction, but it lacks a reasonable explanation for why these specific architectural choices (e.g., using exactly three receptive fields or this particular combination of CNN and Transformer features) are optimal for the task.

4. While the authors conduct several ablation studies, these mostly focus on combinations of self-attention and receptive field features. Other aspects, such as the impact of different receptive field sizes, the effect of hyperparameter tuning on stability and accuracy, and the scalability across additional datasets, are not covered in depth.

5. While the model is termed "lightweight," the paper lacks concrete benchmarks on computational efficiency (e.g., inference time, FLOPs, or memory usage). More comparisons on computational resources would strengthen the argument that LT-FMRF is more efficient than CNNs-based or ViTs-based methods.

6. Some architectural details, particularly within the LT-FMRF module and fusion mechanisms, are described in a challenging way for readers to understand. In addition, it would be beneficial to include more detailed visualizations, such as feature maps at different stages or t-sne visualization over training epochs, to give readers more insight into the model's behavior.

**Questions:**

Please refer to the weaknesses.

---

### Official Review · Reviewer_C2Hr · 2024-10-30

**Soundness:** 2
**Presentation:** 2
**Contribution:** 3
**Rating:** 5
**Confidence:** 4

**Summary:**

In this paper, a lightweight Transformer guided by features from multiple receptive fields is proposed for FSFGIC. The designed LT-FMRF has the capability to manage long-range dependencies and take advantage of different frequency bands. Furthermore, it can be embedded into any given few-shot episodic training mechanisms.

**Strengths:**

The proposed method effectively addresses the issues of long-range dependencies and the insufficiency of few-shot data.

**Weaknesses:**

1. The entire model is divided into multiple modules and is quite complex, so a high-level overview of how the modules connect and interact may be beneficial.
2. The paragraph that begins with "The designed LT-FMRF network contains four modules." has an issue with the expression. The first sentence of this paragraph is somewhat abrupt and does not serve as a good transition. Perhaps the first and second sentences of this paragraph could be swapped while maintaining their original meaning.

**Questions:**

1. In the section on Related Work, it is advised to discuss the key limitations of 1-2 key related works.
2. The paper mentions, "The purpose of this is to maintain convolutional inductive bias, reduce the number of training samples required for ViT training itself, and have the capability to deal with long-range dependencies properly." However, what is the principle behind this? Could you elaborate further?

---

### Official Review · Reviewer_4kAq · 2024-11-01

**Soundness:** 3
**Presentation:** 3
**Contribution:** 2
**Rating:** 5
**Confidence:** 3

**Summary:**

This paper addresses the Few-Shot Fine-Grained Image Classification (FSFGIC) task, which involves utilizing limited training samples to accurately classify images into fine-grained categories. To tackle the challenges of long-range dependency management in CNNs and high-frequency information impairment in vision Transformers (ViTs), the paper proposes an innovative LT-FMRF framework. This framework uses a Convolutional Neural Network (CNN) module to capture local feature information and a self-attention mechanism to handle long-range dependencies. A multi-scale feature fusion strategy is employed to extract global and local features from different receptive fields, and element-wise addition fuses these features to enhance the representation of fine-grained categories. Additionally, end-to-end training is supported, allowing for efficient learning from scratch with limited data.

**Strengths:**

① Extensive experiments on five benchmark datasets demonstrate that LT-FMRF outperforms several existing few-shot learning methods in both 5-way 1-shot and 5-way 5-shot tasks. The results highlight the proposed method's broad applicability and robustness across diverse datasets.
② The use of various feature fusion techniques effectively amplifies inter-class differences, reducing confusion between classes. This is particularly crucial for fine-grained image classification, which often deals with small inter-class variance. The fusion of multi-scale features significantly contributes to improving classification accuracy.

**Weaknesses:**

① The LT-FMRF primarily combines existing Transformers with multi-scale receptive
field features. While this fusion method introduces some changes in architecture, it does not present fundamentally new technical approaches conceptually.
② Although this method is termed a "lightweight" Transformer, the paper does not provide detailed explanations of its advantages in terms of parameter count, computational complexity, and inference speed compared to other models. Particularly in practical applications, lightweight typically means a significant reduction in computational and storage costs, and this aspect lacks detailed comparisons and empirical evidence.

**Questions:**

① Regarding the authors' mention that deeper networks do not necessarily lead to better performance in few-shot fine-grained image classification (FSFGIC), could you further explain the reasons behind this? Specifically, does the performance decline of deep networks result from overfitting, or is it due to the large number of parameters that cannot be effectively optimized under few-shot conditions?
② In the article, you mentioned that self-attention features (SA-FTs) are used to capture long-range dependencies in images, but you also noted that they may not effectively capture local textures. Have you considered modifying the design of the self-attention mechanism to enhance the handling of local information? For example, could you integrate the self-attention module more deeply with convolutional operations or introduce more complex hybrid mechanisms to address this limitation?

---

### Official Review · Reviewer_d7nK · 2024-11-03

**Soundness:** 3
**Presentation:** 1
**Contribution:** 2
**Rating:** 5
**Confidence:** 3

**Summary:**

The paper focuses on developing a new lightweight transformer for few-shot fine-grained image classification task (FSFGIC). The paper modifies the traditional vision transformers from two perspectives: (1) adding multi-scale local information, (2) fuse the local, global feature for increasing inter-class differences. The paper then proposes an LF-FMRF (lightweight transformer guided by features from multiple receptive fields) module, containing a self-attention module and multiple receptive field modules to extract global, multi-scale local features, respectively. The designed network contains four modules. Experiments are conducted on 5 different  FSFGIC datasets.

**Strengths:**

1. The proposed method achieves consistent improvements in diverse FSFGIC datasets.
2. The ablation results verify the effectiveness of the proposed module.

**Weaknesses:**

1. Lack of novelty. The MRF module is the only new module in the proposed LT-FMRF Module, as self-attention based feature tensors can be obtained by the original ViT. Besides, adding multi-scale local features into the ViT is not a new idea.
2. The presentation is poor. The authors only list all the detailed processes of how the network extracts features rather than in an organized way.

**Questions:**

Please try to point out and emphasize the difference between the MRF module in this paper and other multi-scale local features extracted modules in other papers.

---

### Note · Authors · 2024-11-12

I have read and agree with the venue's withdrawal policy on behalf of myself and my co-authors.